# Vignette Research Methodology: An Essential Tool for Quality Improvement Collaboratives

**DOI:** 10.3390/healthcare11010007

**Published:** 2022-12-20

**Authors:** Kurlen S. E. Payton, Jeffrey B. Gould

**Affiliations:** 1Cedars-Sinai Medical Center, Department of Pediatrics, Division of Neonatology, Los Angeles, CA 90048, USA; 2California Perinatal Quality Care Collaborative, Stanford, CA 94305, USA; 3Department of Pediatrics, Division of Neonatology, Stanford University, Stanford, CA 94305, USA

**Keywords:** quality improvement, quality improvement collaborative, factorial vignette, antibiotic stewardship, neonatal intensive care unit, practice variation, neonatology

## Abstract

Variation in patient outcomes among institutions and within institutions is a major problem in healthcare. Some of this variation is due to differences in practice, termed practice variation. Some practice variation is expected due to appropriately personalized care for a given patient. However, some practice variation is due to the individual preference or style of the clinicians. Quality improvement collaboratives are commonly used to disseminate quality care on a wide scale. Practice variation is a notable barrier to any quality improvement effort. A detailed and accurate understanding of practice variation can help optimize the quality improvement efforts. The traditional survey methods do not capture the complex nuances of practice variation. Vignette methods have been shown to accurately measure the actual practice variation and quality of care delivered by clinicians. Vignette methods are cost-effective relative to other methods of measuring quality of care. This review describes our experience and lessons from implementing vignette research methods in quality improvement collaboratives in California neonatal intensive care units. Vignette methodology is an ideal tool to address practice variation in quality improvement collaboratives, actively engage a large number of participants, and support more evidence-based practice to improve outcomes.

## 1. Introduction: How Can Vignette Research Methods Help Address Practice Variation and Support Quality Improvement?

Variable patient outcomes within institutions and among institutions is a major problem in healthcare. Outcomes in some hospitals or units may be excellent, while outcomes in others are below expectations. Unwarranted practice variation in healthcare contributes to variable patient outcomes, increases the risk of harm, and increases healthcare costs [1,2,3,4,5,6,7,8]. Some practice variation may be due to patient mix and other contextual factors. However, a notable proportion of variation in outcomes is due to differences in how care is delivered. At a global level, this is said to be due to differences in practice, termed practice variation. Operationally, practice variation for any given situation depends upon the extent to which clinicians make the best possible decisions for a given patient and circumstance. Some practice variation is expected when appropriately individualizing care for a given patient. However, some practice variation is due to the provider preference or style. Addressing this dichotomy between appropriate and inappropriate variation is vital for quality improvement [8]. An essential task for any quality improvement (QI) initiative is to identify situations where sub-optimal decisions are being made and then implement strategies to improve the decisions. However, a key challenge in planning and implementing any quality improvement initiative is truly understanding the variation and the drivers that influence the variation. Without this information, stakeholders conducting QI projects may be misinformed about the ideal approach to improvement. If a team has accurate and objective information regarding practice variation, it can develop interventions to encourage and sustain optimal decisions that are likely to lead to better outcomes. The California Perinatal Quality Care Collaborative (CPQCC) recently explored the use of the research vignette methodology as a strategy to describe variation and optimize improvement within the neonatal intensive care unit (NICU)-based quality improvement collaboratives (QICs). This review describes our experience and lessons from implementing vignette methods in the context of NICU QICs, but these lessons are relevant to improvement efforts in any field of healthcare.

## 2. What Is Vignette Research Methodology?

Vignette research methodology uses narratives with pragmatic manipulation of case characteristics/variables to explore decisions, beliefs, and/or attitudes of the respondents [9,10,11]. The vignette methods are underutilized in healthcare [10,12]. They are commonly used in the social sciences to describe decisions and behaviors that respondents may exhibit in real-life scenarios [13]. In the field of marketing, vignette methods are called conjoint analysis. Conjoint analysis, for example, aims to describe combinations of preferences for certain characteristics of a product considered for purchase by consumers [14,15]. The analysis describes which set of factors are most desired or most important to the consumer. In healthcare, the aim would be to objectively describe the health care practitioner’s decision preferences. The drivers of these decisions can also be explored with vignette methods. The drivers of the decisions may be vital targets of interventions to reduce variation and sustain improvements over time.

Factorial vignette methods involve adding single or multiple characteristics to the base vignette narrative, while keeping other characteristics the same to isolate the impact of the characteristics of interest. This method distinguishes factorial vignette methods from a traditional survey [16]. This strategy capitalizes on the strengths of experimental methods and traditional survey methods, while minimizing the weaknesses of each approach. Figure 1 displays the structure of a vignette with three variables and a question testing the response to each of the three variables. Figure 2 shows an example of the same vignette structure with details of a preterm infant antibiotic stewardship case vignette. Figure 3 shows a term newborn antibiotic stewardship vignette with results showing the impact of two sepsis risk factors on the proportion of prescribers choosing to start antibiotics. The figure displays how the experimental aspect of isolating the impact of the two sepsis risk factors quantifies the impact of these risk factors.

## 3. What Evidence Supports the Validity and Utility of Vignette Methods in Healthcare?

Vignette methods are considered the most efficient and effective methods of identifying and describing healthcare clinician decisions [10,11,12,17,18]. Peabody et al. studied the relative accuracy of vignettes and chart review. They compared the vignette results to the gold standard of standardized patients in an outpatient adult medicine clinic. They found that the vignettes more closely identified the quality of care compared to the chart review [19,20]. Claims data have also been used to evaluate the quality of care. Converse et al. described how vignettes performed better in assessing quality of care, when compared to claims analysis [10]. Vignettes have been used to assess the quality of care between and within several countries to contrast quality on an international scale [21]. Vignette research methods describe variation and also may support reducing variation, improving care, and reducing costs [22]. Computerized vignette case simulations and gamified practitioner engagement platforms with audit and feedback have been shown to improve the quality of care and reduce costs [23]. Overall, these data suggest that vignettes are a valid and effective method of describing clinician decisions to support optimal healthcare outcomes.

## 4. How Does Vignette Design Impact the Accuracy and Validity of Results?

The validity of a vignette survey is dependent on the quality of the writing, design, structure, and methods used to present the questions in the vignettes [11,24]. The aim is to reduce bias and maximize internal and external validity. The specific methods and strategies used to minimize bias and maximize the validity of vignette methods are beyond the scope of this review. However, this is a vital aspect of vignette development. The thoughtful implementation of a range of vignette design elements can optimize the validity of vignette results [10,11,12,17,24,25,26,27]. Conversely, poorly constructed vignettes may be susceptible to bias and misleading interpretation.

## 5. How Can Vignette Methods Support QICs?

QICs are widely used to improve patient outcomes on a large scale. Overall, QICs tend to yield positive results [28,29]. However, evidence supports many opportunities to improve the effectiveness of QICs [30,31,32]. CPQCC has conducted several statewide multi-center NICU quality improvement collaboratives. Overall, CPQCCs collaboratives have led to notable improvements in infant outcomes on a wide scale in California [33,34,35,36,37,38]. The collaboratives are 18 months long and include California NICUs that voluntarily join to participate. These collaboratives have used the Institute for Health Care Improvement (IHI) model for improvement (www.ihi.org, accessed on 14 December 2022) [39].

Individual participating sites have variable success in meeting the improvement goals during the collaboratives. This is common also in other QICs. The reasons for this are multifactorial. Access to resources, QI culture, QI experience, leadership, staffing, and staff engagement, all may impact effectiveness. Sustained improvements are more likely when a strong evidence base for practices was available, with acceptable guidelines and relatively straightforward aspects of care. Additionally, contextual factors that vary between local participants are expected to be an important barrier or driver of improvement. One important contextual factor is practice variation among providers in a given site. Practice variation can also occur at the site level. We noted that practice variation among individuals within a NICU was repeatedly reported as a primary barrier by the participating site QI teams in our recent antibiotic stewardship collaborative and our very low birth weight infant quality improvement collaboratives. Innovative strategies are needed to improve and optimize the effectiveness of QICs [40].

There are several traditional QI tools that were developed to aid improvement. Statistical process control analysis of time series data helps detect improvement while distinguishing signal from noise. The fishbone, or Ishikawa diagram, helps display and organize cause and effect theories. A Pareto chart helps prioritize the targets of interventions of the causes of the problems to be addressed. Plan-Do-Study-Act (PDSA) cycles help organize the approach to improvement by planning, executing a plan to improve, studying this plan, and then capturing the lessons learned to repeat this process in another cycle. The A3 report provides an initial framework for beginning to examine the problem and organizes all of the QI work in a single document. These strategies provide a helpful structure and address specific problems in QI efforts. QICs frequently are aimed at improving care around areas with a wide variation in practice or outcomes. However, the systematic assessment of individual practice variation is not a commonly used strategy in published QIC studies.

## 6. How and Why Did We Integrate Vignette Methods into the Antibiotic Stewardship QIC?

There is a widespread unwarranted use of antibiotics, and the rates of antibiotic use do not correlate with the rates of proven infection [41,42,43]. Historically, antibiotic use in NICUs evolved with a minimal consideration of the adverse effects. The accepted dogma was “better safe than sorry”. In 2013, some CA NICUs had antibiotic utilization rates (AURs) >50%, reflecting that antibiotics were given in half or more of the patient NICU days [42].

Obstetric antenatal group B streptococcal antibiotic prophylaxis has led to markedly reduced rates of early-onset sepsis in newborns [44,45]. As the rates of early-onset sepsis (EOS) decreased, the evidence and awareness of short- and long-term adverse effects of antibiotics increased. These two inversely changing trends and mandates for antibiotic stewardship implementation led many NICUs to prioritize antibiotic stewardship. However, the optimal strategies to improve and sustain improvement have not been clearly defined.

In response to the high rates of antibiotic use and the wide variation among NICUs in California, CPQCC planned and conducted its first antibiotic stewardship collaborative from 2015 to 2017. This study was the largest multi-center (28 NICUs) NICU antibiotic stewardship collaborative reporting the comprehensive monthly AURs among the participating sites. This stewardship collaborative targeted antibiotic use for suspected early-onset sepsis. The collaborative included 28 California NICUs following the IHI model engaged in monthly webinars working together to improve appropriate antibiotic use based on a set of optimal approaches. However, even among these highly motivated NICUs, many sites struggled to improve or sustain reductions in AUR. Individual prescriber practice variation was noted as an important barrier again. After months of work in the collaborative, many sites were having notable challenges with reducing AURs. The sites reported difficulty obtaining consensus among prescribers, and many sites had not finalized EOS guidelines. Statistical control charts for all sites were analyzed individually. A few sites had evidence of improvement. However, just as many showed brief reductions in AUR and lost the gains. Many did not show any notable improvement. Although practice variation was noted to be widespread, we did not have helpful information about this variation. Towards the end of the collaborative, we decided to implement the vignette study as a supplemental strategy to identify and describe practice variation, thus providing objective data on the “therapeutic locations” where suboptimal decisions were being made and where there may be consensus.

We designed vignettes to explore antibiotic decisions on an individual prescriber level. The results of this study are described in a separate manuscript [46]. The vignettes we developed could be labeled “QI vignettes”. As with traditional vignettes, the vignettes were primarily developed to describe decisions. As a secondary aim, they were also intended to stimulate group discussion and promote individual reflection as well as collective reflection to support more appropriate antibiotic use. This sort of “peer comparison” has been shown to improve outpatient antibiotic prescribing in a randomized control trial comparing several antibiotic stewardship strategies in outpatient pediatrics clinics [47]. Identifying unwarranted variation and intervening with feedback can be effective in reducing variation [48]. Thus, we wanted the vignette results to describe the collective thinking of the prescribers and support practice change at the participating NICUs.

We first considered decisions that had the greatest potential to impact antibiotic exposure in early-onset sepsis cases. We focused on decisions to start antibiotics and decisions to discontinue or continue antibiotics once cultures were negative at around 48 h of life. We then added single or multiple variables (factorial method, as described above) to this base narrative and tested the prescribers’ responses to these variables. We expected this to help us quantify the impact of specific variables on antibiotic decisions.

## 7. What Were the Vignette Results and How Did They Benefit the QIC?

Figure 4 displays the results from the vignette exploring “starting antibiotics”. The results showed actionable stewardship opportunities with the first two cases. Each of the first two cases resulted in 30% of prescribers starting antibiotics. With 70% of prescribers choosing to withhold antibiotics, both cases could be considered reasonable targets of efforts to shift that 30% of prescribers into considering withholding antibiotics. The third case showed 80% of prescribers choosing to start antibiotics in the case of infants with respiratory signs and sepsis risk factors. This relative consensus among the group suggests that this is not a priority area to focus efforts on changing practice.

The first case objectively quantified the proportion of prescribers choosing to start antibiotics based on an elevated C-reactive protein (CRP) level. From an antibiotic stewardship perspective, using CRP in this case is not useful to rule in or rule out infection in most cases of early-onset sepsis. CRP has a high negative predictive value, if it is low. However, it has a very poor positive predictive value because it is elevated in 20% of healthy uninfected newborns [49,50]. Our vignette assessments exploring how prescribers have used and responded to elevated CRP levels from 2015 through 2022 described a longitudinal description of practice variation. In 2018, national guidelines recommended that elevated inflammatory markers should not routinely be used to guide decisions on starting or stopping antibiotics in EOS [51,52]. Our preliminary data during our current antibiotic stewardship collaborative continues to show that prescribers order the CRP in hopes of using it for its negative predictive value, yet respond to elevated CRP values by starting and continuing antibiotics, even with a negative blood culture. This vignette result is relevant for QIC administrators, participating sites, non-participating sites, policymakers, and other stakeholders interested in antibiotic stewardship.

Figure 5 shows histograms displaying the results of the vignette exploring “stopping antibiotics”. The first two cases showed consensus, with 97% and 89% of the prescribers choosing to stop antibiotics at 48 h once blood cultures were noted to be negative. The third case that was written to isolate the impact of the longer duration of respiratory signs resolving at 72 h resulted in nearly a 50/50 split, with 48% of the prescribers choosing to discontinue antibiotics, and 52% choosing to continue them beyond 72 h. This case was the case with the highest variation among the six cases. This case could also be considered an actionable target of stewardship efforts. As mentioned, the initial vignette results allowed structured conversations about drivers of decisions. Discussions among the participating sites revealed that the prescriber’s perspectives regarding “culture-negative sepsis” was driving the variability in this result.

## 8. Discussion: How Do the Vignettes Support the Implementation of Evidence-Based Medicine in QICs?

QICs are intended to improve adherence to evidence-based practices and lead to improved patient outcomes. Practice variation can be a notable barrier. It lies between the evidence and the complex task of implementing optimal processes to improve outcomes. An understanding of practice variation, practice consensus, and drivers of individual decisions is vital to optimize quality improvement. Our experience highlighted how the vignettes can address this gap in the traditional QIC process. The Donabedian model describes QI as a product of improvement within three categories: structure, process, and outcomes [53]. Process measures, outcome measures, and balancing measures provide objective information for assessing the performance during QI efforts. Each of these measures has been described as representing the voice of portions of the improvement process. The process measures are considered the voice of the system of care. The outcome measures are the voice of the patient. The balancing measures monitor the system for unintended consequences. Staff decisions, practice variation, and consensus fall under the structure category in the Donabedian model. The individual staff members of the participating NICUs influence the outcomes through their decisions. The voice of the staff responsible for impacting these measures is not traditionally rigorously measured in QICs. Vignette methods can bring the voice of this vital portion of QI into the open to inform the QIC efforts.

We had not previously explored formal methods of assessing individual decisions and decision drivers of the participating NICU staff. We had overlooked the importance of objectively understanding the decisions of individuals and groups. The first vignette survey allowed us to reach participants that were not directly involved with the QI teams. This objective information was useful for the collaborative administrators and the participating teams. The collaborative administrators must prioritize topics and content to reach the aim of improvement. Vignette data on respondent decisions help inform this process. The participating teams noted that information on individual decisions was vital to guiding discussions and determining where they should focus efforts. After a group meeting among their prescribers, one NICU noted they reached consensus in a particular area of antibiotic management. However, the vignette responses collected after their meeting identified specific areas where the prescribers were not in agreement. This helped this NICUs QI team understand where they should focus further efforts. Informal discussions among clinicians may not accurately capture the complexities of practice variation. The vignettes can identify and describe important practice variation details that may otherwise be misinterpreted or completely overlooked. The vignettes provide unique benefits to QIC administrators and participating QI teams.

Pragmatically developed vignettes may help bridge the gap between the current state of practice and the desired evidence-based practice. Knowledge about practice variation among the group reflects the current state of practice. If consensus guidelines are not available, the vignette results can help describe this current state and support efforts to reduce variation as a first step. If evidence-based guidelines are available, the successful application of these guidelines reflects the desired practice state. Evidence-based guidelines can be used to inform the development of vignettes to specifically target areas where practice is not consistent with the guidelines. We considered the ultimate goal of implementation of evidence-based practice during the vignette development process. This is the ultimate aim of the vignettes. The objective is to accurately describe and use the details of current practice variation to support change, moving closer to the desired evidence-based practice.

Using vignettes methods to foster improvement in QICs is supported by the more recently considered behavior change theory. Traditional behavioral change theories in healthcare focused on the individual as the target of change. The assumption was that the individual’s behavior was a product of his/her attitudes and beliefs. This understandably led to the assumption that working to change the individuals’ attitudes, knowledge, and beliefs would likely lead to a behavior change.

However, in today’s complex healthcare contexts, individual actions are heavily influenced by external factors such as peer influences, social dynamics, medicolegal concerns, and pressures to provide value-based care. Acknowledgement of these external influences should lead us to look beyond the individual toward an alternative approach to behavior change. Targeting change in the attitudes of the collective rather than the individual may be a more effective approach [54,55]. Individuals do not practice in a vacuum conveniently hidden from external influence. Based on this theory, a more fitting approach is to target the understanding and modification of peer group norms and expectations. As a behavioral change tool, the vignettes can objectively identify, display, and monitor these norms. Where indicated, desired evidence-based behaviors can be reinforced, and divergent practices can be discussed from a group perspective. We found that using data visualization of vignette results displaying individual decisions among collective decisions of the group provides a concrete method of implementing this theory-based approach. An individual can view their own decisions alongside the range of their peers’ decisions. The individual decisions can be considered within the collective current state. The desired state of practice can be envisioned as a group working towards a common goal. The question “How should I change my practice and what impact could this have on patient outcomes?” changes and becomes “Imagine the impact on patient outcomes, if we collectively adopt this practice change?”. Vignettes can provide a tangible method of approaching behavior change as an effective way to begin improving outcomes through a more systems-based approach to the implementation of evidence-based guidelines.

Our experience of using vignettes to encourage change is supported by the findings of Zamboni et al. in their systematic review exploring and identifying how context and mechanisms of change benefit the QIC [56]. They listed mechanisms of how QICs influence the participants to follow evidence-based practice. Table 1 pairs vignette functions with these mechanisms of influence. We used the vignette results reported to all participants, graphically displayed the results with data visualization, and used them as an anchor for improvement discussions. This process supports these mechanisms by which the QICs empower the participants to improve.

We are further exploring strategies using vignettes in our current and future QICs. We are planning factorial vignette assessments before, during, and after QICs. Presumably, there are distinct benefits of using vignettes at different time points during QICs. Vignettes could be used well in advance of QICs to help inform the development and structure of a project, including the development of the change package, SMART aims, measurement strategy, and areas of focus. This will allow collaboratives to more effectively meet the specific needs of the participating NICUs. Vignette assessment during a QIC could quantify the effectiveness of collaboratives in a formative way. A second vignette assessment similar to the first could help determine the impact on individual and collective practice. This will allow for prioritizing areas to target. Vignette assessment after the conclusion of a collaborative could be used to engage NICUs that were unable to participate. This would provide a formal process of peer comparison among participants and non-participants. This may accelerate the dissemination of progressive practices to promote change beyond the reach of the initial collaborative.

## 9. Conclusions

Practice variation is a primary barrier to improving patient outcomes. QICs are implemented to improve outcomes on a wide scale. However, systematic assessments of practice variation within QICs are not commonly performed. Factorial vignette methods are an ideal strategy to describe both quality of care and practice variation, engage stakeholders, and promote individual and collective change. We used vignette methods to supplement multicenter QICs in California NICUs. The vignettes provide unique benefits during QIC that cannot be realized with traditional QI methods. Vignette methods should be considered an essential strategy to optimize QI projects and studies.

## Figures and Tables

**Figure 1 healthcare-11-00007-f001:**
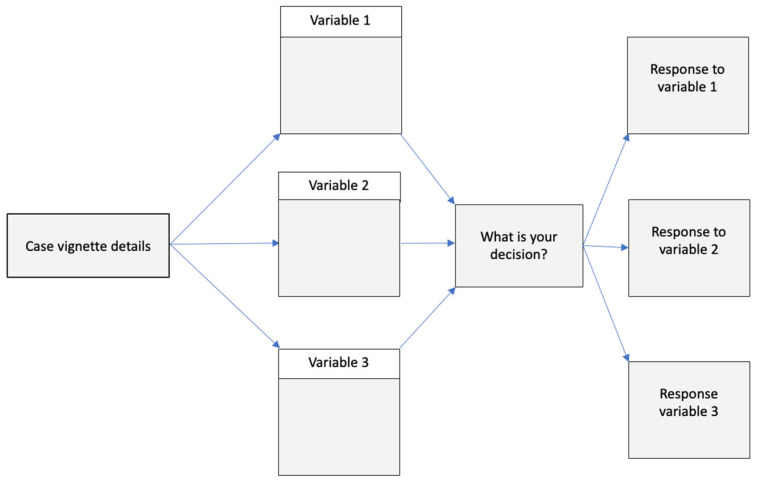
Flow map showing how case vignette details begin the narrative aimed at describing variation. Three variables are each independently added to the narrative to isolate the impact of each variable alone on the decision in response to the variable.

**Figure 2 healthcare-11-00007-f002:**
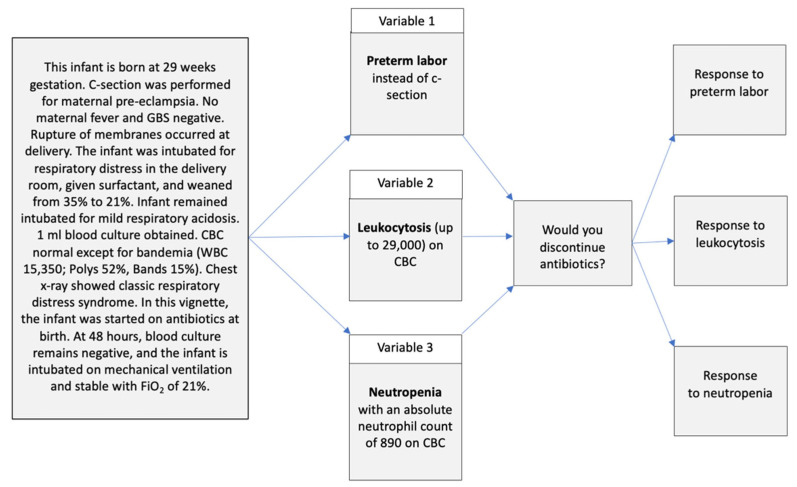
Flow map showing the initial details of a preterm infant case vignette narrative. Prior to the addition of the variables, nearly all respondents chose to discontinue antibiotics. Three variables are each independently added to the narrative to isolate the impact of each variable alone on the decision to discontinue antibiotics.

**Figure 3 healthcare-11-00007-f003:**
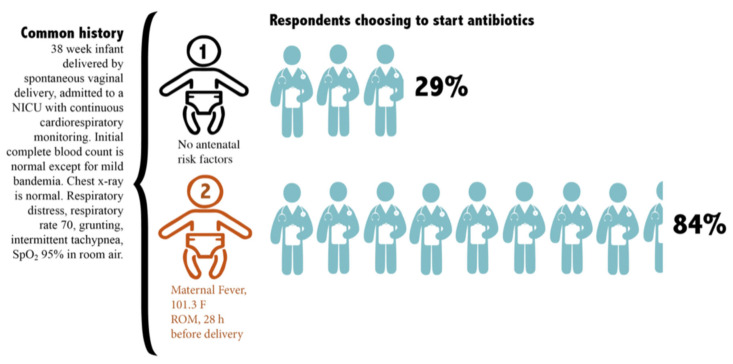
The graphic displays the initial common history of a term newborn vignette. The first variable (**1**) is no antenatal sepsis risk factors. The second variable (**2**) added to the vignette is two antenatal sepsis risk factors. The blue arrow highlights the proportion of prescribers that changed their decision by choosing to start antibiotics in response to the two antenatal sepsis risk factors.

**Figure 4 healthcare-11-00007-f004:**
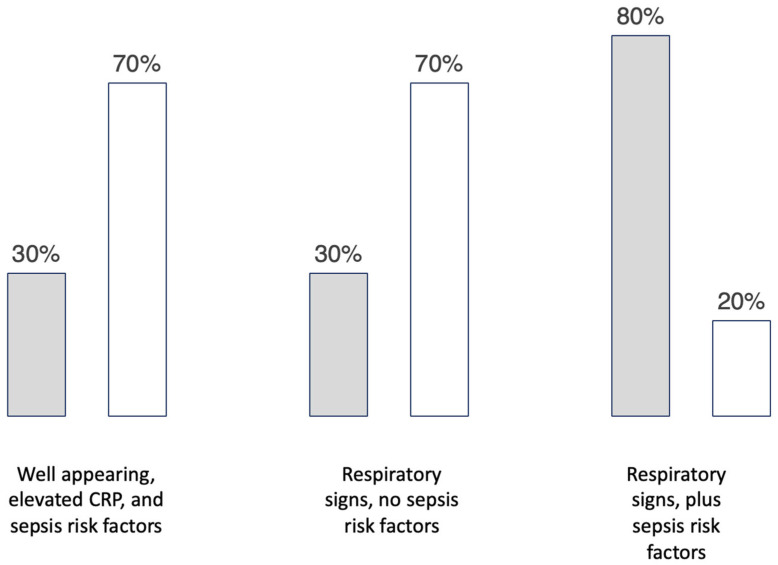
The set of histograms shows the results of a single vignette with three sets of variables added to determine the proportion of prescribers choosing to start antibiotics (grey) and the proportion choosing no antibiotics (white).

**Figure 5 healthcare-11-00007-f005:**
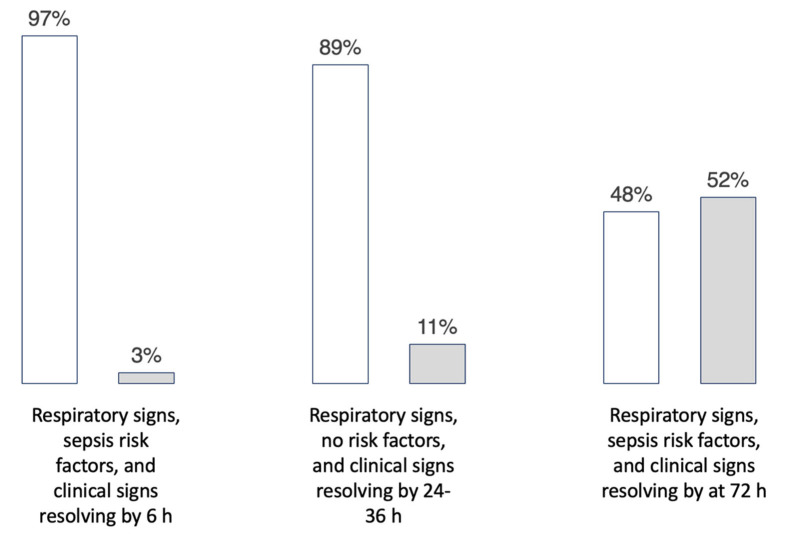
The set of histograms shows the results of a single vignette with three sets of variables added to determine the proportion of prescribers choosing to discontinue antibiotics (white) and the proportion choosing to continue antibiotics (grey).

**Table 1 healthcare-11-00007-t001:** Vignette results support QIC mechanisms of influencing individuals.

How QICs Influence Individuals to Follow Evidence-Based Practice Guidelines	Vignette Results
Increased commitment and confidence in using data to prioritize problems that they can impact	Provide objective data on variation and decisions; Data visualizations of variation can guide discussions about priority areas to target
Increased accountability by making optimal clinical approach very clear	Display range of variation allowing constructive discourse on optimal approaches to improvement
Provide opportunity for peer reflection and group problem solving	Graphical displays comparing proportions of individuals making each decision provide a concrete visualization of opportunities
Bottom-up, inclusive team-oriented shared responsibility; Culture of joint problem solving	Vignettes help reach individuals that may feel distant from the QI work

## Data Availability

Not Applicable.

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
