# Peer review of "Vignette Research Methodology: An Essential Tool for Quality Improvement Collaboratives"

_healthcare, 2022, doi:10.3390/healthcare11010007_

Round 1

Reviewer 1 Report

As a piece of commentary, there arent any issues. Its well written, concise and provides a different aspect to QI/Education that, although done commonly, isnt really named up. 

Author Response

Thank you.

Reviewer 2 Report

The authors make a compelling case for the inclusion of vignette methodology in quality improvement collaboratives to assess and decrease individual caregiver variations in key clinical decisions.  They present real-life, important vignette examples used within a large multi-center collaborative aiming to promote antibiotic stewardship in the neonatal ICU setting.

I am not sure there is enough hard data in this manuscript or even in the publications referenced therein [References 10-12, 17, 18] to state definitively that “Vignette methods are more cost effective relative to other methods to measure quality of care.” (Lines 18-19).  I suggest softening the statement just a little, e.g. “Vignette methods are cost effective relative to other methods…”

Figure 3 on Page 4 is a little confusing because there is no Baby A1; presumably Baby A1 is supposed to be the Common History base case.

On Page 5 reference numeration skips from 40 to 43 and from 45 to 49.

Few minor editing recommendations:

Line 143: Change “pareto” to “Pareto”

Line 165: Consider changing “are” to “were” for more consistent verb tense within paragraph

Line 226: Change “its’” to “its” (no apostrophe)

Lines 306, 319, 342, 399: Change “Jama” to “JAMA”

Lines 307, 367: Change “Bmj” to “BMJ”

Line 366: Change “ALEXANDER CS, BECKER HJ” to “Alexander CS, Becker HJ”

Line 385: Change “Academic pediatrics” to “Academic Pediatrics"

Line 397: Change “Hospital pediatrics” to “Hospital Pediatrics"

Author Response

Response to Reviewer #2

The authors make a compelling case for the inclusion of vignette methodology in quality improvement collaboratives to assess and decrease individual caregiver variations in key clinical decisions.  They present real-life, important vignette examples used within a large multi-center collaborative aiming to promote antibiotic stewardship in the neonatal ICU setting.

I am not sure there is enough hard data in this manuscript or even in the publications referenced therein [References 10-12, 17, 18] to state definitively that “Vignette methods are more cost effective relative to other methods to measure quality of care.” (Lines 18-19).  I suggest softening the statement just a little, e.g. “Vignette methods are cost effective relative to other methods…”

Thank you for the suggestion.  As suggested, we softened our statements about cost effectiveness of vignettes in the abstract and in the body of the manuscript.  The statement now reads, “Vignette methods are cost effective relative to other methods of measuring quality of care.”

Figure 3 on Page 4 is a little confusing because there is no Baby A1; presumably Baby A1 is supposed to be the Common History base case.

Thank you for the suggestion. We labeled the cases “1” and “2” to reflect to clarify that the process starts with case 1 and then the two risk factors are added to create case 2.

On Page 5 reference numeration skips from 40 to 43 and from 45 to 49.

Thank for noting this. We resolved the formatting error and all the references are in order now.

Few minor editing recommendations:

Lines 307, 367: Change “Bmj” to “BMJ”

Line 397: Change “Hospital pediatrics” to “Hospital Pediatrics"

Line 385: Change “Academic pediatrics” to “Academic Pediatrics"

Line 366: Change “ALEXANDER CS, BECKER HJ” to “Alexander CS, Becker HJ”

Lines 306, 319, 342, 399: Change “Jama” to “JAMA”

Line 226: Change “its’” to “its” (no apostrophe)

Line 143: Change “pareto” to “Pareto”

Line 165: Consider changing “are” to “were” for more consistent verb tense within paragraph …

Thank you for the edits.  We revised all of the above in the text.

Reviewer 3 Report

This interesting paper describes a methodological approach useful for quality improvement in medicine. I suggest the authors cover more extensively the relationship between detecting practice variations and applying evidence-based guidelines.
Consensus within a group, especially a small one, does not ensure proper application of evidence-based guidelines. This aspect should be discussed.

Author Response

This interesting paper describes a methodological approach useful for quality improvement in medicine.

I suggest the authors cover more extensively the relationship between detecting practice variations and applying evidence-based guidelines.

Consensus within a group, especially a small one, does not ensure proper application of evidence-based guidelines. This aspect should be discussed.

Thank you for the suggestion. Based on the reviewer’s recommendation, we renamed the discussion section.  The discussion section is now:  “Discussion: how do vignettes support implementation of evidence-based medicine in QICs?”

We revised this entire section to include details of how vignettes can benefit efforts to apply evidence based guidelines.

Reviewer 4 Report

I read with particular interest the manuscript titled “Vignette research methodology: an essential tool for quality improvement collaboratives”. The authors report their experience in applying a vignette methodology to overcome individual barriers to improve antibiotic management in the NICU. They explain very clearly the flowchart they used to choose this method and they explain in a comprehensive and clear way this fascinating new research methodology.

I suggest just a few minor changes:

·      Line 161: The abbreviation EOS has not previously been specified.

·      Line 168: AS is not clear. Abbreviation? 

·      Line 187: A dot is lacking after the word “level”.

In conclusion, I think that the manuscript meets the requirements to be published in Healthcare.

Author Response

I suggest just a few minor changes:

Line 187: A dot is lacking after the word “level”.

Line 168: AS is not clear. Abbreviation? 

Line 161: The abbreviation EOS has not previously been specified.

Thank you for the suggested edits.  We corrected the above issues within the text.